# Emerging Prospects of Nanozymes for Antibacterial and Anticancer Applications

**DOI:** 10.3390/biomedicines10061378

**Published:** 2022-06-10

**Authors:** Nayanika Chakraborty, Sona Gandhi, Rajni Verma, Indrajit Roy

**Affiliations:** 1Department of Chemistry, University of Delhi, Delhi 110007, India; nayanika493@gmail.com (N.C.); gandhi7hd@gmail.com (S.G.); 2Department of Chemistry, Galgotias University, Greater Noida 203201, India; 3School of Physics, Faculty of Science, The University of Melbourne, Parkville, VIC 3010, Australia

**Keywords:** enzyme mimics, nanozyme, reactive oxygen species, antibacterial therapy, anticancer therapy

## Abstract

The ability of some nanoparticles to mimic the activity of certain enzymes paves the way for several attractive biomedical applications which bolster the already impressive arsenal of nanomaterials to combat deadly diseases. A key feature of such ‘nanozymes’ is the duplication of activities of enzymes or classes of enzymes, such as catalase, superoxide dismutase, oxidase, and peroxidase which are known to modulate the oxidative balance of treated cells for facilitating a particular biological process such as cellular apoptosis. Several nanoparticles that include those of metals, metal oxides/sulfides, metal–organic frameworks, carbon-based materials, etc., have shown the ability to behave as one or more of such enzymes. As compared to natural enzymes, these artificial nanozymes are safer, less expensive, and more stable. Moreover, their catalytic activity can be tuned by changing their size, shape, surface properties, etc. In addition, they can also be engineered to demonstrate additional features, such as photoactivated hyperthermia, or be loaded with active agents for multimodal action. Several researchers have explored the nanozyme-mediated oxidative modulation for therapeutic purposes, often in combination with other diagnostic and/or therapeutic modalities, using a single probe. It has been observed that such synergistic action can effectively by-pass the various defense mechanisms adapted by rogue cells such as hypoxia, evasion of immuno-recognition, drug-rejection, etc. The emerging prospects of using several such nanoparticle platforms for the treatment of bacterial infections/diseases and cancer, along with various related challenges and opportunities, are discussed in this review.

## 1. Introduction

Several nanoscale materials are known for their fascinating properties which pave the way for various attractive applications [1,2]. It is well known that due to the high surface-to-volume ratio of nanoparticles, they can function as efficient heterogeneous catalysts [3,4]. This catalytic behavior is also observed in the biological domain, where certain nanoparticles can mimic the function of natural enzymes, which can be attributed to both the intrinsic composition and surface properties of the nanoparticles. These enzyme-mimicking nanoparticles, termed ‘nanozymes’, are structurally different from ‘nanoreactors’, where natural enzymes are simply incorporated/immobilized within nanoparticles [5]. There are several advantages of engineered nanozymes over natural enzymes. The latter are plagued by their inherent proteinaceous character and some intrinsic shortcomings, such as low stability, complex production parameters, high costs and potential immunogenicity, that limit their utilization for combating human diseases [6]. Engineered nanozymes, on the other hand, are non-immunogenic, easy to generate, store, and transport and can operate over a wider range of conditions such as pH, temperature, salt concentration, redox microenvironment, etc. [7,8]. Moreover, the intrinsic physical properties of certain nanoparticles imply that their enzymatic behavior can be tuned using external stimuli such as light, magnetic field, radiation, ultrasound, etc. [9]. However, the specificity and turnover efficacy of natural enzymes are much higher at ambient conditions. Nevertheless, nanozymes have emerged as promising candidates for replacing natural enzymes in several biological pursuits, with the choice between them being reliant on the set of operating conditions available for a specific application.

A variety of nanoparticles that includes those of noble metals, metal oxides/chalcogenides, metal-organic frameworks, carbon-based materials, single-atom-based nanozymes (SANs), etc., are capable of duplicating the action of several key enzymes [10,11]. Apart from low cost, better safety, and high operational stability, a key advantage of nanozymes is the tunability of enzymatic activity by varying their physical parameters, such as size and shape, along with application of external stimuli such as light and magnetic field. For example, Gao et al. in 2007 reported that ferromagnetic oxide (Fe_3_O_4_) nanoparticles had an intrinsic peroxidase-like activity owing to abundant Fe^2+^ and Fe^3+^ ions on their surfaces [12]. The group worked on three different sizes of Fe_3_O_4_ nanoparticles and found that the catalytic activity is size dependent, with smaller-sized particles displaying augmented activity, as the larger surface area provides more binding sites for the substrates.

Nanozymes have functions in all areas where enzymes are required, such as regulation of biomolecular and cellular pathways, cleavage of proteins or polynucleic acids, modulation of oxidative balance, site-specific cleavage of prodrugs, etc. As a result, they have applications in preservation, diagnostics (e.g., in ELISA assays), regenerative medicine, and therapeutics. Indeed, multimodal nanoparticles with intrinsic and/or added functionalities, along with enzymatic action, have emerged as key candidates for numerous attractive biomedical functions. Several authors have reviewed the development of general or specific aspects of nanozyme applications in diagnostics and medicine [13,14,15]. Some key enzymes or classes of enzymes, such as superoxide dismutase (SOD), catalase (CAT), oxidase (OXD), peroxidase (POD), deoxyribonuclease (DNase), etc., have been implicated in both antibacterial and anticancer therapy. Enzymes that catalyse Fenton-type reactions are also included in this category [10,11,12]. Such enzymes and the general reactions that they are known to catalyse are depicted in Figure 1. The objective of this review is to provide an account of the various nanoparticles that are known to mimic the function of such enzymes and thus serve as antibacterial and anticancer agents. We shall particularly focus on the latest developments and underline the future prospects of this area. Owing to the vast body of reports already available on this topic, most of which have been published recently, we are unable to comprehensively cover this area; rather, we will highlight selected examples from the same. We shall also discuss the various challenges and opportunities that include the safety and clinical prospects related to the use of nanozymes in antibacterial and anticancer therapy.

## 2. Nanozyme Platforms for Antibacterial Applications

Bacterial diseases are a major public health concern worldwide [16,17]. Although antibiotic treatment is the most widely recognized paradigm for treating such diseases, the continuous use, overuse, and misuse of antibiotic-based drugs over a long time have prompted the evolution of multi-drug-resistant super-bacteria [18,19]. Drug resistance is further promoted since bacteria thrive predominantly in surface attached extracellular polymeric substance (EPS) matrix-enclosed-multicellular communities or as biofilms. Drug-resistant bacterial diseases currently account for the loss of 700,000 lives annually. Unless novel antibiotic-free interventional strategies are developed urgently, the mortality rate is anticipated to skyrocket to 10 million lives annually by 2050 [20]. 

To overcome this challenge, enormous efforts have been focused on the discovery and development of alternative broad-spectrum antibacterial agents or strategies [21,22]. In this respect, antibacterial peptides, bacteriophage therapy, and antibacterial enzymes have gained popularity over recent years. In addition, nanomaterial-based antibacterial platforms such as photothermal therapy (PTT) [23], photodynamic therapy (PDT) [24], and photocatalytic therapy (PCT) [25] are also becoming prevalent. Recent studies have confirmed that the excellent activity and high substrate selectivity of certain endogenous enzymes, such as natural POD and OXD, can restrain the escalation of bacterial growth and disrupt biofilms. These enzymes catalyse critical biochemical reactions to locally generate toxic reactive oxygen species (ROS) that oxidize key bacterial components such as cell membrane/wall or intracellular compartments. However, owing to the various shortcomings of natural enzymes, engineered nanozymes are being extensively used as promising alternatives in antibiotic-free antibacterial therapy [26]. Nanozymes are also less likely to develop bacterial resistance owing to good membrane permeability and biocompatibility [27]. More importantly, nanozymes can be equipped with catalytic activities to eradicate bacterial biofilms [28,29,30]. In the subsequent section, we have provided selected examples of antibacterial nanozymes, which are roughly classified based on their composition.

### 2.1. Metal-Based Nanozymes

Several noble-metal- (gold, silver, platinum, etc.) based nanozymes display strong catalytic activity. Zheng et al. developed mercaptopyrimidine conjugated Au nanoclusters (Au NCs) targeting intractable superbugs in vitro and in vivo. The positive charge of the nanozyme facilitated their easy adherence on the bacterial surface and subsequent cell membrane damage. The induction of intracellular ROS production in bacterial cells was mainly accredited to the intrinsic oxidase-like and peroxidase-like activity, leading to killing about 99% of bacteria and promoting the wound healing process [31]. Zhang et al. evaluated bimetallic platinum–copper (PtCu) alloy nanoparticles for both peroxidase-like and ferroxidase-like in a weakly acidic medium and detection of Fe^2+^, in addition to their antibacterial potency [32]. Similarly, Cai et al. formulated core–shell Pd@Ir bimetallic nanostructures using a seed-mediated-growth method with morphology-dependent bactericidal activity. This study established that the Pd@Ir octahedron showed better antibacterial activity when compared to Pd@Ir cubes due to higher OXD-like activity. The oxidation of 3,3′,5,5′-tetramethylbenzidine using these Pd@Ir nano cubes had a 1.7 times higher V_max_ value and a 4.4 times higher V_max_ value for Pd@Ir nano octahedron than when catalyzed by Pd cubes. Moreover, the study also revealed that the OXD-like activity of Pd@Ir became elevated in the presence of natural organic matter. Upon interaction with humic acid (HA), the nanozyme generated high levels of ROS and also promoted cellular internalization of the nanostructure [33].

The high coordination capability of the Cu ion with amino acids prompted the development of Cu-based nanozymes with intrinsic POD-like activity. Cu-embedded hydrogel-based nanozymes can provide effective coverage of wounds and accelerate the wound-healing process with the assistance of H_2_O_2_ by stimulating angiogenesis and collagen deposition [34]. As mentioned previously, the nanozyme activity can be enhanced using external stimuli, such as light, magnetic field, and electricity. Karim et al. demonstrated that upon photoactivation, the POD-like activity shown by CuO nanorods are significantly enhanced owing to their favorable band structure. In the presence of H_2_O_2_, about 20 times enhanced production of the ROS was observed upon light illumination of the nanorods, leading to potent physical damage to bacterial cells treated with the nanorods [35]. Bimetallic CuCo_2_S_4_ nanoparticles have demonstrated exceptional POD-like activity and high antibacterial capability at neutral pH, when compared to monometallic CuS and CoS NPs. These bimetallic nanozymes could accelerate the healing of burn injuries infected with methicillin-resistant *S. aureus* (MRSA) in vivo [36]. 

### 2.2. Metal Oxide/Sulfide-Based Nanozymes

Cerium oxide nanoparticles (CeO_2_) are a classic example of biological catalyst with high POD-like activity attributed to a reversible redox switch between Ce^4+^ and Ce^3+^ ions. A CeO_–_H_2_O_2_ system is more conducive to promote ROS due to very high and efficient POD-like activity. Different shapes and sizes of nanoceria lead to multiple enzymatic activities, such as that of SOD, CAT, POD, and OXD, because of surface-rich oxygen vacancies, smooth oxygen diffusion, and high redox potential [37]. Luo et al. developed an imidazolium-type poly (ionic liquid) (PIL)/cerium(IV) ion-based electrospun nanofibrous membrane (PIL-Ce) exhibiting DNase mimetic catalytic activity and rapid wound healing in an MRSA-infected mice model. The high antibacterial potential of PIL-Ce was evaluated, and the disintegration of resistant genes was also investigated to block the spread of drug resistance [38]. 

*S. Enteritidis* is a notorious zoonotic food-borne pathogen that evades antibiotic treatment and thrives within host cells in animals. An iron oxide nanozyme (IONzyme) with intrinsic POD-like activity was formulated, which can catalyze H_2_O_2_ to generate **·**OH radicals under acidic conditions for suppressing the survival of intracellular *S. Enteritidis* [39]. Following the co-localization of IONzymes with *S. Enteritidis* in acid autophagic vacuoles of leg-horn male hepatoma-derived cells, ROS-regulated antibacterial action against intracellular *S. Enteritidis* was observed via autophagic elimination (Figure 2). Furthermore, transcriptomic profiling displayed that IONzymes changed the hepatic oxidation reduction along with autophagy-related gene expression in chicken livers infected with *S. Enteritidis*.

Biofilms are a complex consortium of bacterial communities embedded within a self-produced exopolysaccharide extracellular matrix that create a localized and protected microenvironment adhered to the surface. In a recent study by Gao et al., nano-iron sulphide particles were synthesized by the conversion of a natural organo-sulphur compound derived from garlic with broad-spectrum antibacterial activity against resistant bacterial infections. Nano-iron sulphide is a nanozyme exhibiting POD-like and CAT-like activities, which catalyzes the oxidation of H_2_O_2_ to accelerate the production of highly toxic hydrogen polysulphide and expedite 500-fold antibacterial efficacy against drug-resistant bacteria. This nanozyme also has the potential to fight biofilms on human dental caries and accelerate wound healing [40]. The bacterial strain *S. mutans* accumulates on the tooth surface utilizing sugar-rich acidic conditions and demineralizes the enamel-apatite causing caries biofilms. Catalytic Fe_3_O_4_ nanoparticles are shown to target and degrade the caries-causing biofilms with high specificity under acidic conditions without impacting surrounding oral tissues in vivo. These nanoparticles generate **·**OH radicals at low concentrations of H_2_O_2_ (1%), disrupting the embedded *S. mutans* in the protective biofilm matrix. Furthermore, the dissolution of hydroxyapatite in the acidic conditions was reduced when the caries was treated with the nanoparticles, providing a mechanistic insight in the prevention of dental caries [41]. 

In another example, it was shown that dextran-coated iron oxide nanoparticles (Dex-NZM) exhibited 49% co-localization with bacteria and 51% with the exopolysaccharides while avoiding binding to gingival cells. Dextran coating concurrently increased the stability, the selectivity catalytic activity, and the treatment efficiency of Dex-NZM nanoparticles against the biofilm matrix due to the ease of incorporation into the extracellular matrix of the biofilm, thus enhancing the matrix breakdown [42]. A ferritin nanozyme (Fenozyme) was developed by the integration of recombinant human ferritin protein shells (HFn) specifically targeting the blood–brain barrier endothelial cells with an inner Fe_3_O_4_ nanozyme core exhibiting CAT-like activity. Fenozyme possesses the capability to protect the integrity of the blood–brain barrier and prevent cerebral malaria as was observed in an experimental cerebral malaria mouse model [43]. Fe-based nanozymes possess POD, CAT, OXD, glucose oxidase (GOx), sulphite oxidase (SOx), and SOD-like activity, with the potential to treat biofilm-associated diseases and combat bacteria. 

### 2.3. Carbon-Based Nanozymes

Carbon-based nanomaterials, such as carbon nanotubes (CNTs), Carbon dots (CDs), graphene and its derivatives, carbon nitride, and fullerene have been widely applied in the field of biomedicine owing to their physiochemical properties, biocompatibility, and multi-enzyme-mimicking activities. Their superior mechanical properties also facilitate their use as dressing materials for the healing of infected wounds. Wang et al. prepared a series of carbon nanotubes with rich oxidized groups (o-CNTs), exhibiting superior POD-like activity over a wide pH range [44]. The carbonyl group on the surface of o-CNTs acted as active catalytic centers, whereas the carboxyl and hydroxyl groups presented competitive sites. Due to an inherent negative charge and the tendency to form hydrogen bonding, the carboxyl group has a higher inhibitory tendency on the catalytic propensity than the hydroxyl group. Therefore, 2-bromo-1-phenylethanone-modified o-CNTs (o-CNTs-BrPE) were prepared to reduce the inhibitory effect of the carboxyl group in the nanozyme. As the number of competing sites decreased, o-CNTs-BrPE exhibited good POD-like activity, thus enabling catalysis of H_2_O_2_ to **·**OH and leading to the eradication of bacteria and the protection of tissues from bacteria-induced edema and purulent inflammation. 

Graphene quantum dots (GQDs) display superior enzyme mimetic activity owing to rapid electron transfer when compared to graphene oxide. Chen et al. developed a GQD-based antibacterial nanozyme system capable of generating highly toxic ROS that showed enhanced and broad-spectrum antibacterial activity. Moreover, GQD-based band-aids exhibited exceptional in vivo wound healing at low concentrations of H_2_O_2_ without conceding cell proliferation [26]. In another example, hybrid nanozymes were constructed by incorporating Au NPs within graphitic carbon nitride nanosheets (g-C_3_N_4_ NSs). The incorporation of AuNPs increased the POD-like activity and robust antibacterial potency against multidrug resistant *S. aureus* due to synergism between the nanosheets and the AuNPs [45]. 

Carbon nanosheet-based stimuli-responsive biomaterials have been developed recently to treat bacterial infections. Ren’s group developed copper ion-doped carbon nanosheets (Cu-SA) to behave as POD and glutathione (GSH) peroxidase mimics. Glutathione is part of the defense mechanism of bacteria that protects it from oxidation by scavenging excess ROS from the bacterial cells. This hybrid nanozyme was incorporated within a biocompatible and thermo-responsive hybrid nanogel made up of bacterial cellulose nanowhiskers (BCNWs) and poly(N-isopropylacrylamide). The resulting Cu-SA@BCNW/PNI hybrid nanogel showed high antibacterial action via the generation of excess ROS at the cost of intracellular GSH (see Figure 3). This intelligent hybrid nanogel has a remarkable sol–gel transition response at human physiological temperature. This ability allows the hybrid nanogel to form a gel over infected regions in situ for faster disinfection of wound sites [46].

### 2.4. Transition Metal Dichalcogenide (TMDC)-Based Nanozymes

A number of 2D transition metal dichalcogenides (TMDCs) have also emerged as promising antibacterial agents owing to their large 2D surface area and intrinsic enzyme-like properties. Recently, a defect-rich adhesive MoS_2_/rGO vertical heterostructure (VHS) was synthesized, with abundant elemental vacancies and a rough surface that facilitates better bacterial capture and ROS production through local topological interactions. The nanozyme exhibited exceptional antibacterial activity because of triple enzymatic (POD-like, CAT-like, and OXD-like) activity and surface defects [47]. Another group has reported that in comparison to MoS_2_ nanozymes, defect-rich MoS_2_ flower-shaped nanozymes have superior antibacterial efficacy owing to two reasons. First, the rough surface of the latter promotes higher adhesion to bacterial cells. Second, their defect-rich active edges facilitate improved POD-like activity owing to thermodynamically favorable lower adsorption energy of substrate H_2_O_2_ and desorption energy of **·**OH radicals [48].

The light-absorbing capability of TMDC-based nanoparticles can be exploited for photoactivated enhancement of enzymatic activity. Shan et al. reported that Cu_2_MoS_4_ nanozymes can be activated by irradiation with near infrared light (1064 nm, 1 W cm^−1^). This photoactivation could enhance the OXD and POD-like activity of the nanozymes, leading to remarkable anti-bacterial activity against 8 log MDR *E. coli* and 6 log *S. aureus* [49]. The activity of the Cu_2_MoS_4_ nanozymes can be further tuned with topographic modifications. On the other hand, Niu et al. developed an intelligent photoregulated strain-selective bactericidal strategy based on a charge tunable MoS_2_ nanozyme and a photoactive molecule. The nanozyme was synthesized by citraconic anhydride-modified PEI-MoS_2_ and the charge tuning was achieved by adjusting the pH responsive citric anhydride. This system has the capability of being modulated by light for charge reversal on the surface and concurrently leading to the enzymatic activation of MoS_2_ upon varying the pH of the system [50].

Silver nanoparticles (Ag NPs) display a wide-spectrum antibacterial propensity attributed to the release of Ag^+^ ions that causes structural deformity and protein inactivation of bacterial cells. Recent reports have substantiated that Ag NPs can mimic enzymes such as SOD, CAT, OXD, and POD. The activation of molecular oxygen on the surface of Ag NPs leads to the enzymatic activity. A hybrid Fe_3_O_4_@MoS_2_-Ag nanozyme was constructed by a hydrothermal method and in situ photo deposition of Ag NPs on a defect-rich rough surface of MoS_2_ to capture the bacteria. The nanozyme resulted in the inactivation of ~69.4% *E. coli* due to the synergistic disinfection caused by the POD-like activity, the leakage of Ag^+^ ions, and the NIR (808 nm) light-activated photothermal effect resulting in the production of ROS (see Figure 4). In addition, the magnetic property of Fe_3_O_4_ was exploited to recycle the nanozyme [51].

### 2.5. Prussian Blue (PB) and Metal–Organic Framework (MOF)-Based Nanozymes 

Biocompatible Prussian blue (PB) nanocrystals have shown moderate levels of antibacterial activity through POD and OXD-mimetic generation of ROS in bacterial cells. However, as PB nanocrystals generate heating upon exposure to visible light, their antibacterial activity can be significantly enhanced upon combined photothermal therapy. We have demonstrated that the antibacterial action of the PB nanocrystal platform can be further enhanced upon coating with a layer of chitosan that facilitates high affinity electrostatic interaction with both Gram-positive and Gram-negative bacterial cells [52]. In a separate work, we have shown that upon doping silver ions in the PB network, very high broad-spectrum antibacterial action could be observed due to the combined effects of ROS generation and silver-ion-mediated toxicity, along with a photothermal effect [53]. 

Metal organic frameworks (MOFs) are hybrid nanomaterials comprised of organic bridging links and metal nodes, with unique 3D structures having distinctive features, such as specific pore size, diversity in porous structure, and large surface area. Biocompatible MOFs not only serve as hosts for stabilizing natural enzymes, but also serve as catalytic sites with high enzyme-like activities. The high degree of order in the framework provides proper arrangement of active catalytic sites, leading to better interactions with the substrates [54,55]. Ren et al. demonstrated a nature-inspired strategy for the construction of an MOF-based functional enzyme mimic with active sites engineered in a pore microenvironment and pseudopodia-like surface to enhance the bacteria-trapping capability [56]. This nanozyme serves as a faithful POD mimic, with its metal nodes acting as the active centers and nanosized cavities as binding pockets. This system is engineered in such a way that the microenvironment around the active site enriches and activates the substrate molecule for bacterial inhibition. The MOF-nanozyme displayed enhanced antibacterial efficacy as its pseudopodia-like surface promoted effective trapping of bacteria. In another study, an Au-doped MOF/Ce-based nanozyme (MOF-2.5Au-Ce) was designed with DNase- and POD-mimicking activities to disrupt biofilms [57]. The cerium(IV) complexes act as DNase mimics, which hydrolyze the extracellular DNA (eDNA) and components of already manifested biofilms. In addition, the MOF displays strong POD-like catalytic activity against the bacteria rendered in the biofilm, thus controlling the recurrence of biofilm or recolonization of bacteria (Figure 5). 

### 2.6. Single Atom Nanozymes (SANs) 

Recently, single atom nanozymes (SANs) have emerged as unique catalytic agents for a variety of applications. As compared to ‘conventional’ nanozymes, where irregular active site distribution or truncated surface densities lead to reduced catalytic activity and specificity, in SANs, the active sites are maximized due to then even distribution of metal centers [9]. Various SANs, such as Pt–Cu, Pt/CeO_2_, M–N_5_ and M–N_4_ (M = Fe, Co, Zn, etc.), have been developed [58,59,60]. These SANs display various enzyme-mimicking properties, such as POD, SOD, CAT, and glutathione peroxidase (GPx-like) activities, and have great potential in anti-inflammation, anti-bacteria, therapeutic diagnosis and degradation of organic pollutants [61,62,63]. Shi et al. synthesized single iron atom nano-catalysts by anchoring single iron atoms in N-doped amorphous carbon (SAF NCs) which induces POD-like activity and is highly effective in bacterial peroxidation. These SAF NCs have shown excellent antibacterial activity against both Gram-positive *S. aureus* and Gram-negative *E. coli* bacterial cells, with a low MIC of 62.5 μgmL^−1^ in vitro. Furthermore, the antibacterial activity can be significantly enhanced by 808 nm NIR laser irradiation. Combined with the good photothermal property of SAF NCs, in vivo bacterial infections can be effectively treated, resulting in better wound healing [64].

Liu et al. synthesized a zinc-based zeolitic-imidazolate framework (ZIF-8)-derived carbon nanomaterial containing atomically dispersed zinc atoms with an efficient POD-like activity. This high activity of this SAN is attributed to the presence of coordinatively unsaturated Zn–N_4_ active sites which has been confirmed by density functional theory (DFT) calculations. In the presence of H_2_O_2_, this SAN inhibited the growth of *P. aeruginosa* by up to 99.87% and also promoted in vivo bacteria-infected wound disinfection and healing [65]. Recently, SANs with carbon nanoframe-confined FeN_5_ active centers, which catalytically behaved like the axial ligand-coordinated scheme of cytochrome P450, were reported by Huang et al. Synergistic effects furnish FeN_5_ SANs with a clear electron push-effect mechanism and also the highest OXD-like activity among other nanozymes (the rate constant 70 times higher than commercial Pt/C) and versatile antibacterial applications [66].

## 3. Nanozyme Platforms for Anticancer Applications

Cancer is a medical condition which is characterized by the uncontrolled growth of malignant cells, leading to solid localized tumors followed by extensive spread (metastasis) throughout the body. It is to date one of the leading causes of death all over the world. The conventional approaches for cancer treatment include chemotherapy, radiotherapy, and surgery. Though these are being widely practiced to mitigate cancer in humans, they suffer several shortcomings. One of the major problems is the non-specificity of these techniques which ends up harming the normal tissues. In addition, drug resistance and increased toxicity in the case of chemotherapy is also a worrisome challenge [67,68]. Therefore, newer approaches are required for eliminating both solid tumors and metastatic spread without localized or systemic toxicity in the body. In this respect, multimodal nanoparticles offering unprecedented diagnostic and therapeutic abilities have emerged as promising platforms for combating cancer [2,69]. 

The immediate vicinity of tumor cells (tumor microenvironment, or TME) is mildly acidic and accompanied by the overproduction of H_2_O_2_, low oxygenation (hypoxia), and reduced activity of certain enzymes such as CAT [70,71]. These aspects contribute to the nourishment, growth, and metastatic spread of cancer cells and foster their evasion from immuno-degradation [72,73]. Nanoparticle-mediated modulation of the TME, therefore, has emerged as an attractive option for the treatment of cancer. The enzymatic action of nanoparticles can lead to significant altering of the TME through various known or perceived mechanisms that include catalyzing Fenton-type reactions, cleaving prodrugs to locally release toxic drugs, degrading tumor-promoting oncogenes, duplicating the action of enzymes such as CAT, etc. [74,75,76]. Such nanozyme-mediated TME modulation can lead either to direct killing of cancer cells or complement other therapies such as chemotherapy, photothermal therapy (PTT), and photodynamic therapy (PDT). 

Chemo-dynamic therapy (CDT) is a very recent concept that is characterized by the in situ activation of Fenton or Fenton-like reactions to locally decompose endogenous H_2_O_2_ and generate ROS, such as hydroxyl (**·**OH) and superoxide (**·**OOH) radicals at tumor sites [77,78]. The locally generated ROS triggers the death of cancer cells via oxidative-stress-induced degradation of intracellular components such as proteins, lipids, and nucleic acids. Moreover, since a Fenton reaction is considerably quenched by a low supply of H_2_O_2_ and a neutral microenvironment, this approach has little effect on normal cells/tissues. Several nanoparticles, such as metals, metal oxides, carbon-based, and metal–organic frameworks, are known to promote CDT by duplicating the action of Fenton-reaction activating enzymes [79,80,81]. Therefore, targeting such nanoparticles to the TME can lead to the robust killing of cancer cells without harming surrounding healthy cells. 

In addition to the direct triggering of cancer cell death, the enzymatic action of certain nanoparticles can complement other cancer therapeutic modalities. For example, photodynamic therapy (PDT) of cancer relies heavily on an ample supply of molecular oxygen in the tumor region. Therefore, the inherent hypoxia associated with most tumors severely hampers the efficacy of PDT [82]. Tumor hypoxia can be substantially relieved by the conversion of endogenous H_2_O_2_ into molecular oxygen; however, the enzyme CAT, which promotes this conversion, is also poorly present in the TME. Several nanoparticles, such as those of noble metals such as gold and platinum, are known to duplicate the action of CAT and locally generate high concentrations of molecular oxygen. Thus, targeting such nanoparticles in the TME can significantly enhance the efficacy of concurrently administered PDT for enhanced therapeutic action [83]. PDT drugs, or photosensitizers, can also be incorporated with these nanoparticles and targeted to cancer cells/tissues to ensure maximum therapeutic gain. Overall, it has been observed that several therapies such as chemotherapy, PDT, and PTT can function synergistically with CDT, with each supporting the other in a synergistic, cooperative manner. For example, it was reported that by increasing the TME temperature via PTT, the rate of the Fenton reaction was also increased [84]. In the subsequent section, we have provided selected examples of anticancer nanozymes, which are roughly classified based on their composition.

### 3.1. Metal and Metal Oxide/Sulfide-Based Nanozymes 

Jiang et al. formulated a hybrid nanozyme, composed of melanin-coated manganese dioxide nanoparticles and ultrasmall gold nanoparticles (MMF-Au). They showed that these nanozymes could be used for low H_2_O_2_ tumors for enhanced CDT. MMF-Au induced consecutive reactions of glucose oxidation and a Fenton-like reaction. The enhanced efficacy of CDT was confirmed by in vitro and in vivo experiments [85]. Fu et al. synthesized biomimetic CoO@AuPt nanozymes by assembling Au and Pt nanoparticles onto the surface of hollow CoO nanostructures. These could trigger a series of reactions to produce substantial ROS without any stimuli. They observed specific tumor damage in both in vitro and in vivo experiments, which proved these nanozymes to be highly efficient in tumor therapy [86]. Hybrid nanozymes can also be designed for cascade catalytic reactions. In an example, Jing et al. formulated a bimetallic iron–copper nanozyme (FeCu PNzyme) and then co-encapsulated it with the natural enzyme GOx and the anticancer drug mitoxantrone (MTO) [87]. The resulting formulation, termed FeCu-GOx PNzyme-MTO, was biocompatible and acted as a carrier and as a therapeutic agent. Cascade catalytic therapy was demonstrated by a GOx-induced conversion of intratumoral gluconic acid to H_2_O_2_, which in turn was converted to ROS by the FeCu nanozyme. In addition, owing to the inherent photothermal conversion ability of the FeCu nanoformulation, light activation resulted in thermally induced triggering of mitoxantrone release, leading to a potent therapeutic effect. In mice bearing a 4T1 tumor, a combination of FeCu-GOx PNzyme-MTO treatment and light activation led to a significant inhibition of tumor growth (Figure 6).

Even magnetically induced therapy can be combined with nanozyme-induced chemodynamic therapy. Both of these are non-invasive and are not restricted by depth limitations. Iridium-ion-doped manganese ferrite nanoparticles were fabricated for efficient cancer therapy. The magnetic nanoparticles produced heat upon exposure to alternating magnetic field. Iron reduced by glutathione was responsible for CDT, and iridium ions were used to target the mitochondria. The increased temperature enhanced the efficacy of CDT, and the disturbance in the cellular redox homeostasis induced by CDT made cells more sensitive to magnetic field. Synergism was observed with the help of in vitro and in vivo experiments [88].

CDT has worked in synergism in yet another non-conventional therapy: sonodynamic therapy (SDT). SDT is also a localized treatment which utilizes ultrasound radiation to generate singlet oxygen which kills the cancerous cells. This technique is non-invasive and yields deep tissue penetration. However, it suffers from decreased efficacy due to tumor hypoxia and insufficient generation of singlet oxygen. To overcome this limitation, a technique combining SDT with CDT was tried. For this purpose, metal organic frameworks (porous coordination network-224 or PCN-224) containing platinum nanozymes were prepared and then loaded with doxorubicin (Dox). The overall formulation (DOX@PCN-224/Pt nanoparticles) was successful in converting H_2_O_2_ to hydroxyl radicals and oxygen, and alleviated hypoxia. The enhanced availability of molecular oxygen considerably improved the efficacy of SDT; it also enhanced the chemotherapeutic effect of doxorubicin as observed with the help of in vitro and mice experiments (Figure 7) [89]. In a similar study, PtCu_3_ nanocages were used which exhibited enhanced SDT due to efficient CDT. This formulation had dual enzyme-like activity, horseradish peroxidase and glutathione peroxidase. Therefore, in addition to producing oxygen and ROS, it is also responsible for depletion of the glutathione present in the tumor cells. This depletion of glutathione helps in the enhancement of SDT, as confirmed by in vitro and in vivo experiments. The formulation showed extremely low toxicity to the normal cells and also had potential of dual modal imaging [90].

Wei et al. combined the effects of starvation therapy of cancer with CDT. They formulated IrRu-GOx@PEG nanostructures, IrRu alloy nanoparticles were coated with PEG, and then it was loaded with the enzyme GOx. IrRu nanozymes were responsible for the catalytic conversion of H_2_O_2_ to carry out CDT, and PEG enhanced the bioavailability of the nanoformulation. GOx was responsible for converting glucose to H_2_O_2_ and cut off the nutrient supply of the tumor (starvation therapy). Since starvation therapy is limited due to hypoxia in tumor cells, CDT produced oxygen to eliminate this problem, which ensured continuity of the starvation-therapy-related reactions. The formulation effectively caused apoptosis in the 4T1 cancer cell line and effectively treated breast cancer as observed with the help of in vivo results [91]. 

Nanozymes fabricated out of cerium oxides have gained much attention lately for their impressive biomedical applications. This could be attributed to their very peculiar autocatalytic properties. Shi et al. synthesized an SOD-mimicking nanozyme based on cerium oxide, CeO_2_-Gd. The formulation combines the fluorescent properties of gadolinium with the antioxidant properties of CeO_2_ nanoparticles. In vitro studies, using BGC-803 cells, showed that higher SOD mimetic activity was observed when the ratio of Ce^3+^/Ce^4+^ was higher in the nanozymes. The doping of a rare earth metal, gadolinium, generated oxygen vacancies in the nanozyme. This enhanced the fluorescence and antioxidant properties, as observed in the BGC-803 cells [92]. Another interesting class of nanomaterials that has found application as nanozymes is magnetic iron oxide nanoparticles (MIONs). They can show pH-dependent enzymatic activity; at mildly acidic pH, they can use H_2_O_2_ as a substrate and break it down to produce highly toxic reactive oxygen species, hence exhibiting peroxidase-like activity. Mansur et al. made a hybrid nanoformulation by conjugating magnetic iron oxide with GOx and then coating it with a shell of a biocompatible polymer: carboxymethyl cellulose. The structure turned out to be supramolecular with vesicles. The formulation has dual enzymatic activity: the POD-like activity of iron oxide and GOx. The formulation was successful in killing brain cancer cells as confirmed by in vitro studies using U-87 MG cells. In addition, the nanozyme-induced cell death was more prominent in cancer cells compared to normal cells due to their lower pH [93]. 

Recently, cancer immunotherapy has emerged as a promising treatment paradigm. However, there is a limitation to its efficacy for cancer treatment due to the immunosuppressive properties of the TME. In a recent report, Y. Wei et al. employed bio-orthogonal chemistry for the reprogramming of TME. They fabricated a mannose vector decorated palladium bio-orthogonal nanozyme. This formulation could synthesize (in situ) vorinostat, which is a histone deacetylase inhibitor that has the ability to alter the TME (Figure 8). It was observed that the nanozyme was preferentially becoming accumulated inside the M2 macrophages which helped to avoid inflammation of the normal tissues. The success/advantage of this formulation was in the synergistic effect of the remodeling of TME and peroxidase-like activity. In colon-cancer-bearing mice, the nanozyme effectively targeted M2 macrophages, which in turn activated the immune system and inhibited the tumor growth considerably [94]. 

Radiotherapy is still widely used in combination with surgery and chemotherapy to mitigate cancer. However, there are challenges in its usage: inefficient accumulation of radiation energy inside the tumor and resistance to radiation due to hypoxia. Li et al. fabricated porous platinum nanozymes to overcome these challenges. The nanozyme has a high Z-element and the capacity to generate oxygen; so it can considerably enhance DNA damage (induced by radiation) and oxidative stress. The formulation could effectively deposit X-ray energy inside the tumor cells and cause the cell cycle to arrest. It could also generate oxygen efficiently by using H_2_O_2_, hence exhibiting peroxidase-like activity. It was used to synergistically enhance radiotherapy to kill tumor cells. In vivo studies also showed that the platinum nanozyme was safe to use; it did not cause any apparent toxicity [95].

### 3.2. Carbon-Based Nanozymes 

In a recent study, self-assembled nanoparticles were formulated using copper ions, carbon dots, and doxorubicin (anticancer drug). The particles released the active components in the acidic tumor environment. Doxorubicin increased the level of H_2_O_2_ in cells, which in turn enhanced the chemodynamic effect produced by the CAT-like activity of carbon dots and copper ions. It was shown with the help of in vivo experiments that the nanoparticles were very highly biocompatible and had synergistic anti-tumor activity [96]. T. Luo et al. fabricated tetramodal nanoparticles made from iron-doped carbon dots (Fe@CDs) attached with DNA for combining gene therapy with PTT, CDT, and sensing (see Figure 9). The nanozyme could efficiently catalyse the Fenton reaction for CDT, which could be significantly (six fold) enhanced upon NIR light-induced heat generation (photothermal effect) in an animal model. Similarly, there was a two-fold enhancement in the efficacy of gene therapy. Overall, the synergistic therapy resulted in excellent inhibition in tumor growth and 80% survival for 50 days in a mice model of 4T1 breast tumor [97].

It has been observed lately that intratumor bacteria impart resistance of tumors against gemcitabine (a chemotherapy drug). The bacteria inside the tumor express an enzyme, cytidine deaminase, which converts gemcitabine to its inactive form. This resistance can be nullified by inhibiting the cytidine deaminase activity of the intratumor bacteria. Xi et al. prepared carbon nanospheres doped with nitrogen (N-CSs) to work on the gemcitabine resistance. The formulation had the potential to show dual functionality; it could inhibit the activity of cytidine deaminase and function as nanozymes for tumor catalytic therapy. Nitrogen doping imparted POD-like activity, which produced **·**OH radicals to carry out chemodynamic therapy. The graphite-like structure doped with nitrogen can bind to the active site of the enzyme competitively and thus prevent the breakdown of gemcitabine. It was indeed observed in the mice model that N-CSs could effectively nullify gemcitabine resistance imparted by intratumor bacteria. In addition to this, there was synergism in the cancer therapy involving nanoenzymatic activity for chemodynamic therapy and gemcitabine for chemotherapy [98].

### 3.3. Other Nanozymes 

Sheng and colleagues encapsulated the photosensitizer chlorin e6 (Ce6) within MIL-100 metal organic frameworks and tagged hyaluronic acid (HA) on the MOF surface. The MOF behaved as the enzyme POD and promoted Fenton-reaction-mediated generation of hydroxyl radicals and molecular oxygen. While hydroxyl radicals triggered CDT, enhanced oxygen production facilitated Ce6-mediated PDT following light activation, thus resulting in cascade photo-chemo dynamic therapy of cancer. Furthermore, HA promoted tumor-specific delivery [99]. In another example, Ce6 was loaded within a single atom nanozyme composed of a single ruthenium atom anchored on a support of Mn_3_[Co(CN)_6_]_2_ MOF. The single ruthenium atom (active catalytic site) duplicated the action of the enzyme CAT that led to higher oxygen production for relieving tumor hypoxia and thus promoting PDT of cancer [100]. Recently, Li et al. developed a multifunctional nanoprobe based on PEGylated MoSe_2_/Au nanospheres. While the POD- and CAT-mimicking activities of MoSe_2_ promoted CDT via the conversion of tumor-rich H_2_O_2_ into hydroxyl radicals and molecular oxygen, Au could be activated with NIR light resulting in a very high photothermal conversion efficiency of 73%. This combination of CDT and PTT on primary tumors in mice could trigger cancer immunotherapy that promises to eliminate both primary and metastatic tumors [101]. In a recent demonstration, a nanozyme (Yb-Pb@PDA) was formulated by doping ytterbium ions in Prussian blue nanoparticles and then coated with polydopamine. The combined effect of an enhanced Fenton reaction was observed aided by NIR light irradiation, resulting in an appreciable antitumor effect in vivo [102].

A summary of the various antibacterial and anticancer applications of nanozymes covered in this review is provided in Table 1.

## 4. Challenges and Opportunities of Nanozymes

**Hurdles for clinical approval of nanozyme:** More than 923 nanozymes have been reported through nearly 15 years since the inception of the idea of enzyme mimics. These bionic nanomaterials with complex composition and tunable physicochemical properties equip nanozymes with flexibility, versatility, and adjustability to practical applications in the industry. Currently, nanozymes with activities such as peroxidase, oxidase, catalase, phosphatase, etc., are utilized as delivery vehicles and have the capability to trigger the immune system and improve the thermostability for bio-distribution and availability. However, as an emerging field, there are a lot of challenges for nanozyme-based disease diagnosis and translation to an industrial scale.

The enzyme-mimicking ability of nanozymes is limited. Oxidoreductases (e.g., peroxidase) and hydrolases (e.g., phosphatase) are widely applied, but the utilization of other classes of enzymes are rarely studied and researched. The scope of application can only be broadened with an expansion of the types of nanozyme activity.Disease diagnosis and monitoring require high specificity for biomolecular sensing. It is imperative to improve nanozyme substrate specificity for better interaction. During enzymatic catalysis, the change in the orientation of molecules also plays an important role, and understandings of deep mechanisms are crucial for the development of highly specific nanozymes. Decoding the exact molecular mechanism for multi-enzyme mimicking with regard to electron movements is still not well-understood. The catalytic activity of nanozymes needs to be improved. Though a small section of nanozymes enjoy higher catalytic activity than natural enzymes, most nanozymes fall short in comparison. Successful bioconjugation of nanozymes is chiefly affected by their low catalytic activity. Molecular imprinted polymers (MPI) on nanozymes enhance both specificity and catalytic activity of enzymes by polymerizing the binding sites on the substrates. Substantial efforts are required to industrialize and streamline the bioconjugation protocol for large-scale production. In addition, SANs is also an attractive strategy to improve the enzyme-like activity because of atomic utilization efficiency and highly dispersed catalytic active sites. Ultrasensitive detection of analytes is possible through SERS-based sensing techniques. SERS-based nanozymes can improve detection sensitivity. There are not many research reports highlighting nanozyme-enhanced SERS techniques in the field of biosensing. In addition, nanozyme-based lateral flow assay (LFA) is a realistic approach for biosensing. To date, only a few studies report on integration of nanozymes in LFA-based biosensors.To broaden the scope of application of nanozymes, it is essential to exhibit high catalytic activity at a wider range of pH and temperature. This will not only enhance the versatility of nanozymes but also encourage utilization of nanozymes for disease detection and treatment.Studies have indicated that most of these nanozymes are safe for use in animals due to their analogy in structure and composition with the FDA-approved nanoparticles [103]. However, the safety of noble-metal nanozymes is a gap because the toxicity and bio-capacity of noble-metal nanozymes while being used as a biocatalyst or in vivo cell protection or cancer treatment is a problem [104]. Although there are numerous reports on a cytoprotective role and the biocompatibility of nanozymes, metal overload leads to side impacts on normal tissues. Nanoceria has a ROS-eliminating property and protects the cell from ROS damage. Biocompatibility and physiological stability offer a great challenge towards development of future clinical prospects. The thorough understanding of toxicity and encouraging the development of low-toxic material is essential. Overall, nanozymes should be extensively evaluated for their bioavailability, ADME (absorption, distribution, metabolism, excretion), as well as short-term and long-term toxicity [105].Clinical samples contain various complex biological matrices which can be adsorbed nonspecifically on the surface of nanozymes and lead to false-positive results. Signal transduction reduces the specificity of a biosensor because of the aggregation of nanozymes. The nanozyme solution needs to be kept away from an environment rich in ROS and ultraviolet sources.Future breakthroughs in this rapidly developing field of nanozyme research with large-scale applications in clinical translation still need to be realized to make nanozymes beneficial for human disease diagnosis and survival. It is important to promote nanozymes in nanozymological techniques. Nanotechnology, computational chemistry, and artificial intelligence can augment the nanozyme enzymatic potential.In biomedical applications, the final goal is the translation of new probes in treating human diseases. Even though clinical research involving nanozymes is still in its infancy, there are several indications that these artificial enzymes are more promising than their natural counterparts for the clinical treatment of cancer and bacterial diseases. For example, since iron-oxide-based nanoparticles have already been used as clinical MR contrast agents, iron-oxide -based nanozymes can be considered as promising candidates in this regard. On the other hand, Prussian blue (PB) nanoparticles are used clinically for the removal of radioactive contaminants in humans [52,53]; thus, PB-based nanozymes can also be explored for advanced clinical aspects. Other nanozymes based on biocompatible nanoparticles, such as graphene oxide, metal–organic frameworks, single-atom nanozymes, etc., are also being extensively explored in this regard. 

In our understanding, nanozymes will have high possibilities of potential applications which span from in vitro detection and in vivo monitoring and catalytic therapy in the near future. The dynamics of nanozymes with regards to pharmacokinetics, absorption, metabolism, bio-distribution, therapeutic duration, excretion, and toxicity of nanozymes needs to be studied and understood at various stages of administration and treatment. Nanozymes without targeting molecules tend to accumulate in the liver, lung, and spleen according to some bio-distribution reports. Progression in clinical application of nanozymes primarily will depend upon biocompatibility and therapeutic safety.

## 5. Conclusions

From the various reports covered in this review, it is amply clear that the enzymatic properties displayed by several nanosized particles have widespread applications in, among other disciplines, both antibacterial and anticancer therapies. As both these therapeutic approaches are based on killing infected/rogue cells, the mechanisms of action of the nanozymes are similar, mostly relying on the generation of toxic ROS and/or overcoming hypoxia through modulation of various biologically relevant redox reactions and pathways. In addition, it is evident that the nanozymatic therapeutic action synergises with other anticancer/antibacterial therapies, such as chemotherapy, radiation therapy, PDT, PTT, etc., leading to potent combination therapy and/or theranostic modalities.

Another exciting prospect of nanozymes is their ability to show multienzymatic behavior, which can be exploited to drive cascade catalytic reactions [11]. For example, the glucose oxidase-like action of a nanozyme converts endogenous glucose into peroxide, which in turn is decomposed into cytotoxic ROS via the peroxidase-like action of the same nanozyme. However, multienzymatic action can often be a roadblock for therapy, wherein contrasting enzymatic action can lead to poor substrate affinity and catalytic action. Therefore, efforts are also underway to investigate the effect of various: (a) nanoparticle parameters, such as their size, shape, and composition (e.g., Fe^2+^/Fe^3+^ ratio in iron oxide nanoparticles); (b) reaction parameters, such as pH, temperature, redox environment; and (c) external stimuli, such as the application of light, magnetic field, radiation, etc., for tuning the action of nanozymes to suit a particular application. Optimization of such parameters will lead to higher specificity and efficacy of the nanozymes.

As evident from the reports covered in this review, most of the therapeutic nanozymes function by altering the oxidative balance/pathways of the body. In addition, future research should also focus on developing nanomaterials for duplicating the action of other enzymes which are relevant in cancer and bacterial diseases such as asparaginase, glutaminase, lysozyme, etc. In addition, more research effort should be invested in nanozymatic reversal of drug resistance mechanisms, such as by showing cytidine deaminase-like behavior. 

For the application of nanozymes in the body, it is critical to evaluate their safety aspects, biocompatibility, biodegradability, excretion, etc. Most of the nanozymes reported in these applications, such as metal nanoparticles, metal–organic frameworks, etc., have a decent safety profile. In particular, single atom nanozymes are extremely promising owing to their ultrasmall dimensions and ease of excretion. Nevertheless, detailed investigations about their biosafety profile are critical for their widespread use. 

With regard to the antibacterial action of nanozymes, in addition to their application in the living body, they can also be employed for sanitization of large surfaces in densely populated areas, such as schools, hospitals, railway stations, stadiums, etc. Therefore, efforts are underway for the stable incorporation of nanozymes on fabrics, metals, woods, etc., which are frequently touched by people. Another important aspect is sanitization of drinking water, which can be facilitated by nanozymes stably loaded on filters. Antibacterial nanozymes can also be applied in the packaging, storage, and transportation of food items, medicines, etc. 

## Figures and Tables

**Figure 1 biomedicines-10-01378-f001:**
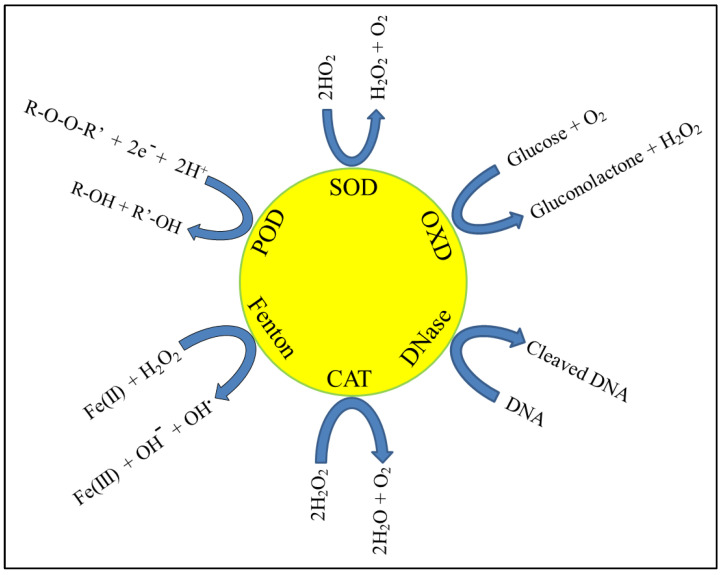
Key enzymes and their reactions for antibacterial and anticancer use.

**Figure 2 biomedicines-10-01378-f002:**
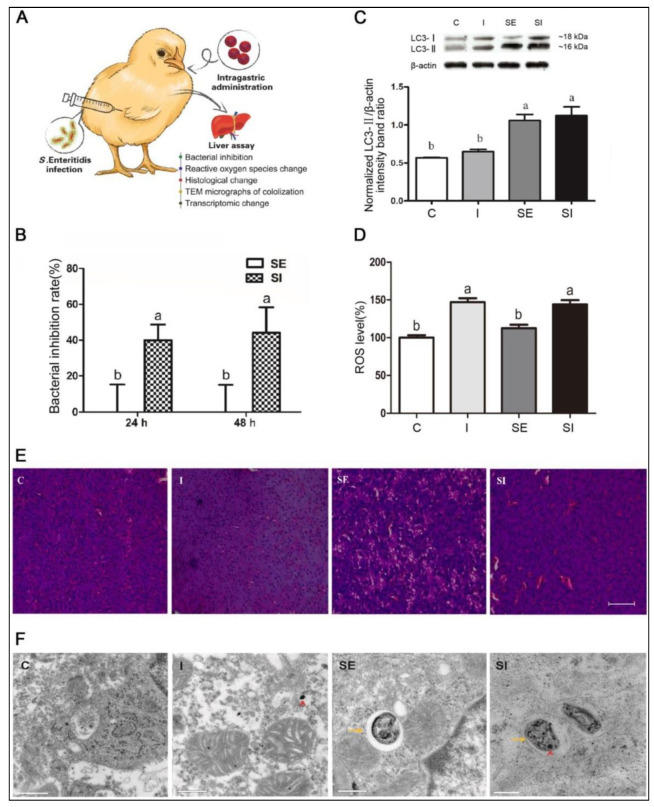
Iron-oxide nanozyme (IONzyme) with peroxidase-like activity in the treatment of chickens infected with *S. Enteritidis*. Study groups: vehicle control (**C**); IONzyme alone (I); *S. Enteritidis* without (SE) or with (SI) IONzyme: (**A**) schematic illustration of the experiment in chickens, which were subcutaneously administered with the bacteria and orally administered with IONzymes; (**B**) bacterial inhibition in infected livers, without and with IONzyme treatment, at 24 and 48 h post-treatment; values represent the mean ± SEM (*n* = 9). Different letters indicate statistically significant difference (*p* < 0.05).; (**C**) Western blot analysis showing normalized hepatic LC3 protein expression levels for the various study groups; the values represent the mean ± SEM (*n* = 3). Different letters indicate statistically significant difference (*p* < 0.05); (**D**) the relative levels of ROS in various study groups; values represent the mean ± SEM (*n* = 6). Different letters indicate statistically significant difference (*p* < 0.05); (**E**) photomicrographs of the liver histological sections for the various study groups; scale bar: 50 µm; (**F**) TEM micrographs showing bacterial co-localization with IONzymes within autophagosomes of chicken liver. IONzyme: red short triangle arrow; S. Enteritidis: yellow arrows. Scale bar: 0.5 µm (in C, I, SE) and 0.2 µm (SI). Reprinted with permission from Ref. [39]. Copyright 2018 IVSPRING.

**Figure 3 biomedicines-10-01378-f003:**
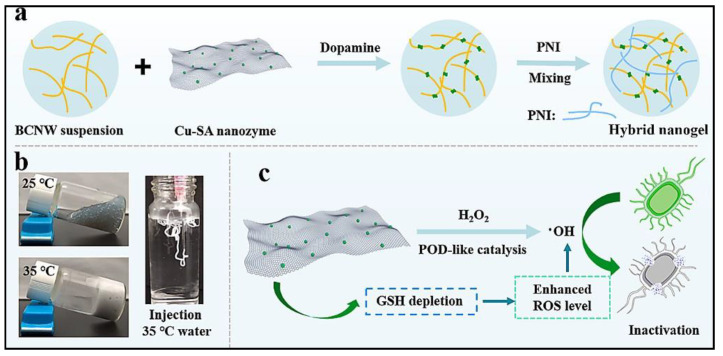
Schematic representation showing: (**a**) fabricating of Cu-SA@BCNW/PNI hybrid nanogels from BCNW (bacterial cellulose nanowhiskers), Cu-SA nanozyme, dopamine, and PNI (poly-N-isopropylacrylamide); (**b**) thermo-responsiveness of Cu-SA@BCNW/PNI hybrid nanogels; and (**c**) mechanism of antibacterial activity of the nanozyme. Reprinted with permission from Ref. [46]. Copyright 2022 Elsevier.

**Figure 4 biomedicines-10-01378-f004:**
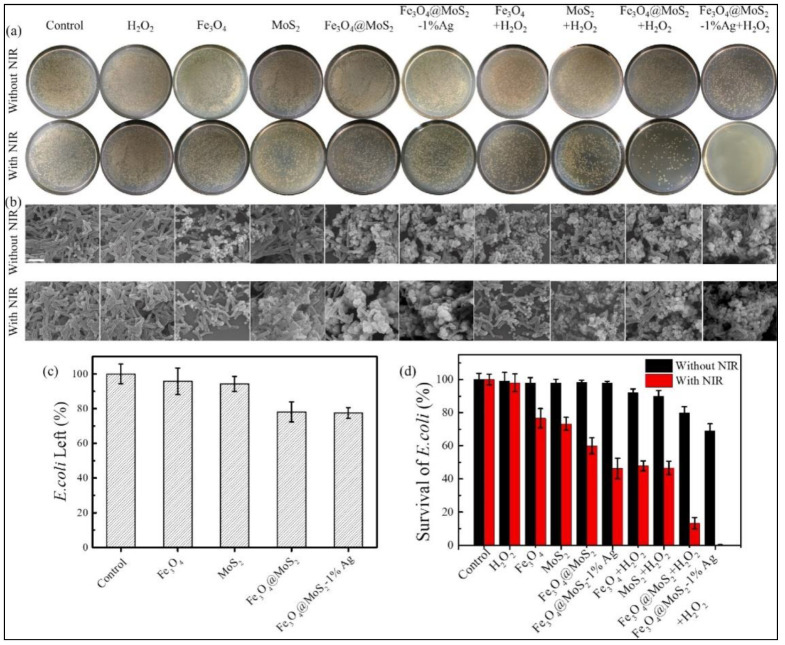
(**a**) Photographs of standard agar plates showing viable bacterial colonies for various controls and samples, without and with NIR light irradiation; (**b**) SEM images of bacteria treated with corresponding samples; (**c**) bacteria percentage left in the suspension after removing the catalysts; (**d**) percentage survival of bacteria treated with different controls and samples. Reprinted with permission from Ref. [51]. Copyright 2021 Elsevier.

**Figure 5 biomedicines-10-01378-f005:**
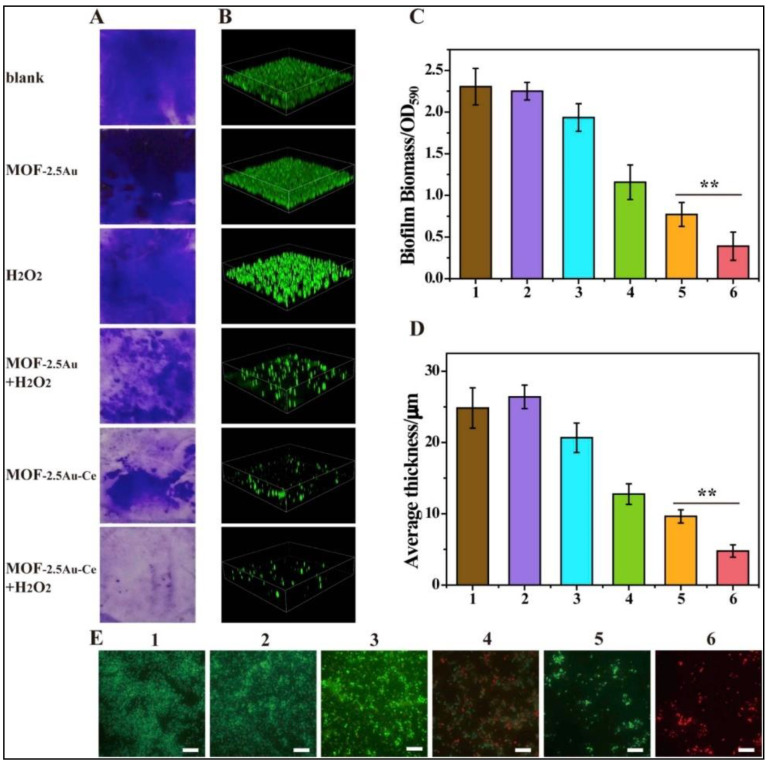
MOF-2.5Au-Ce nanozymes for inhibiting bacterial biofilm formation. (**A**,**C**) Crystal violet staining, and (**B**,**D**) 3D Confocal Laser Scanning Microscopic (CLSM) imaging of biofilms treated with various samples for 24 h. Optical density at 590 nm (OD_590_) was measured to quantify the biomass of biofilms. Error bars were calculated on the basis of three independent experiments; ** represents *p* ≤ 0.01. Image sizes of CLSM: 315 μm × 315 μm. Biofilm thickness was quantified with Comstat 2 software. (**E**) LIVE/DEAD stain images of residual biofilms using fluorescence microscopy. Green and red stains indicated live and dead bacteria, respectively. Scale bar = 10 μm. Various treatment samples: (1): control (culture medium only); (2): MOF-2.5Au; (3): H_2_O_2_; (4): MOF-2.5Au + H_2_O_2_; (5): MOF-2.5Au-Ce; and (6): MOF-2.5Au-Ce + H_2_O_2_. Reprinted with permission from Ref. [57]. Copyright 2019 Elsevier.

**Figure 6 biomedicines-10-01378-f006:**
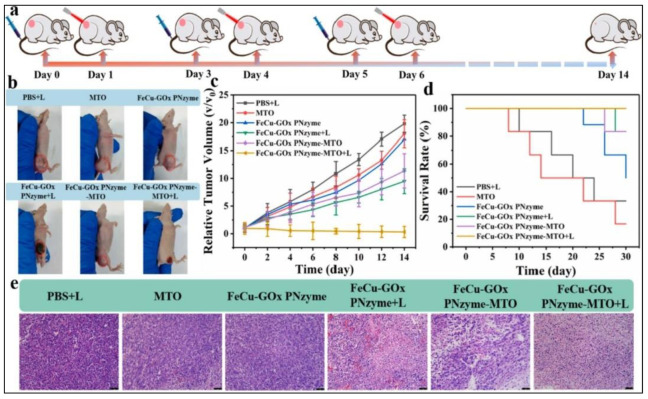
In vivo antitumor effect in mice bearing a 4T1 tumor, treated with (1) PBS + Light; (2) MTO; (3) FeCu-GOx PNzyme; (4) FeCu-GOx PNzyme + Light; (5) FeCu-GOx PNzyme-MT; and (6) FeCu-GOx PNzyme-MTO + Light. The data was shown as means ± SD (*n* = 5): (**a**) schematic illustration of the in vivo experimental schedule; (**b**) photographs of mice and tumors from various mice groups post-treatment; (**c**) time-dependent tumor growth curves; (**d**) Kaplan–Meier diagram of time-dependent cumulative surviving profiles of various study groups; (**e**) hematoxylin and eosin (H and E)-stained images of tumor slices collected from various study groups of tumor-bearing mice. Scale bar: 75 μm. Reprinted with permission from Ref. [87]. Copyright 2022 Elsevier.

**Figure 7 biomedicines-10-01378-f007:**
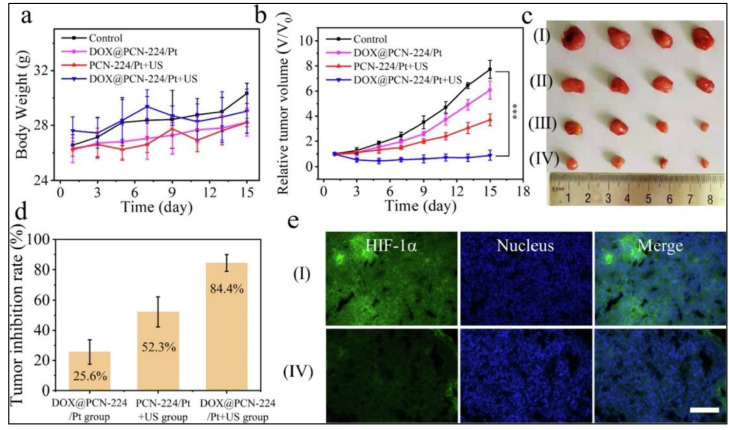
(**a**) Body weight and (**b**) relative tumor volume variation of mice from different study groups (*n* = 4, *** *p* < 0.001 (highly significant); (**c**) photos of excised tumors on Day 15 post-treatment (I: Control; II: DOX@PCN-224/Pt; III: PCN-224/Pt + US; IV: DOX@PCN-224/Pt + US; US: Ultrasound); (**d**) tumor inhibition percentages of various groups (*n* = 4); (**e**) effect of DOX@PCN-224/Pt nanoparticles on tumor hypoxia and HIF-1a expression: Immunofluorescence images of tumors following treatment with saline (control) or DOX@PCN-224/Pt + Ultrasound. (I: Control; IV: DOX@PCN-224/Pt + US; US: Ultrasound) (scale bar is 50 μm). Reprinted with permission from Ref. [89]. Copyright 2022 Elsevier.

**Figure 8 biomedicines-10-01378-f008:**
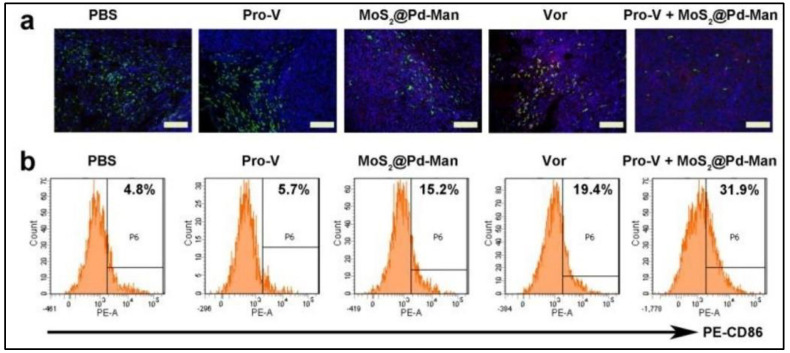
In vivo reprogramming of TME: (**a**) immunofluorescence images showing the tumor-distribution of M1 and M2 macrophages (red: M1U; green: M2U); Scale bars: 100 μm; (**b**) flow cytometry analysis of the expression levels of CD86 in F4/80+ macrophages within the tumor tissuesPE-CD86: anti CD-86 antibody; various study groups: PBS (control); Vor (vorinostat); Pro-v (vorinostat prodrug); MoS_2_@Pd-Man (mannose vector decorated palladium bio-orthogonal nanozyme); and Pro-V + MoS_2_@Pd-Man. Reprinted with permission from Ref. [94]. Copyright 2022 Elsevier.

**Figure 9 biomedicines-10-01378-f009:**
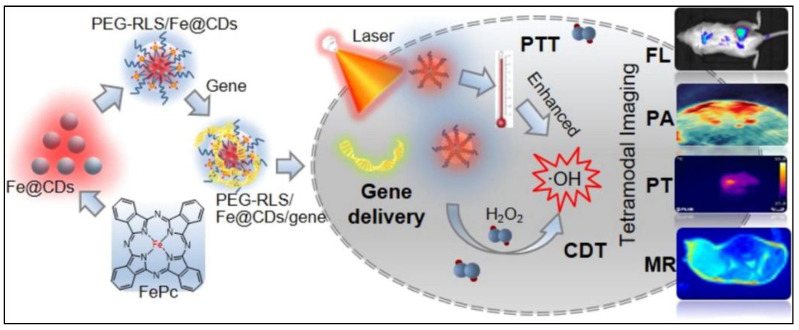
Schematic representation of the formation of the multimodal theranostic nanoparticles based on iron-doped carbon dots (Fe@CDs) as the photoactivated nanozyme. The gene-loaded nanoparticles were capable of tetramodal imaging (FL: fluorescence; PA: photoacoustic; PT: photothermal; and MR: magnetic resonance), along with laser-light-activated photothermal therapy (PTT) and a Fenton-reaction-driven chemodynamic therapy (CDT), for a potent antitumor effect. Reprinted with permission from Ref. [97]. Copyright 2021 Elsevier.

**Table 1 biomedicines-10-01378-t001:** A tabular summary of the use of various nanozymes in antibacterial and anticancer applications covered in this review (NA: data not available).

Nanozyme Formulation	Mode of Action	Particle Size (nm)	Zeta Pot. (mv)	Ref.
**Metal-based nanozymes**				
Mercaptopyrimidine conjugated Au NCs	Intrinsic peroxidase-like and oxidase-like activity for the generation of intracellular ROS in bacterial cells.	TEM: <2 DLS: 1.8 ± 0.7	+37.6 ± 1.1	[31]
PtCu NPs	Both ferroxidase- and peroxidase-like activity for antibacterial applications.	DLS: ~44 ± 3.4	NA	[32]
Core-shell Pd@Ir bimetallic nanomaterials with an ultrathin shell	High oxidase-like activity and morphology-dependent antibacterial activity	TEM: 14 DLS: NA	NA	[33]
Copper and amino acid containing hydrogel-based artificial enzymes	Peroxidase-like activity for fighting wound pathogens with accelerated wound healing by collagen deposition and angiogenesis	FESEM: 50–70 DLS: NA	NA	[34]
Bimetallic FeCu nanozymes, co-encapsulated with GOx and mitoxantrone	Cascade catalysis of intratumoral glucose to gluconic acid and H_2_O_2_, followed by production of ROS in TME.	TEM: 45 × 14 DLS: 65	+ 42.7	[87]
Platinum nanozymes co-encapsulated with Doxorubicin within MOFs	Catalysed the conversion of H_2_O_2_ to hydroxyl radicals and oxygen, for synergizing CDT with SDT	TEM: 100 DLS: 143	−10.6	[89]
PtCu_3_ nanocages	SDT efficacy enhancement via ROS generation and glutathione depletion by behaving as both horseradish peroxidase and glutathione peroxidase.	TEM: 14.29 DLS: 34	−6.28	[90]
GOx loaded and PEG coated IrRu alloy nanozymes.	Peroxidase-mimicking conversion of H_2_O_2_ for triggering CDT, along with Gox-mediated promotion of starvation therapy of cancer by depleting intracellular glucose.	TEM: 43 DLS: NA	NA	[91]
Mannose-decorated Pd bioorthogonal nanozyme	Peroxidase-mimicking in situ production of the immunostimulatory agent vorinostat for the promotion of cancer immunotherapy.	TEM: 40 DLS: NA	−20	[94]
Porous Pt nanozyme	Promotion of cancer radiotherapy via the peroxidase-mimicking generation of ROS	TEM: 65.09 DLS: 115.6	−3.84	[95]
**Metal-oxide/sulphide-based nanozymes**				
CuO nanorods	Peroxidase-mimicking generation of ROS from H_2_O_2,_ further facilitated by irradiation with visible light.	TEM: 70.1 ± 14.7 DLS: NA	NA	[35]
Bimetallic CuCo_2_S_4_ nanoparticles	Enhanced peroxidase-like activity at neutral pH. Healing of burn wounds infected with drug-resistant bacteria via generation of highly toxic hydroxyl radicals.	TEM: 30 DLS: 68	NA	[36]
FeS nanoparticles with polysulfanes	Enzyme-like activity for accelerating the release of polysulfanes, for potent bactericidal activity against pathogenic dental biofilms.	TEM: 20–30 DLS: NA	NA	[40]
Iron oxide nanoparticles (IONzymes).	Catalysis of hepatic oxidation–reduction and autophagic gene regulation pathways.	TEM: 200 DLS: 350	NA	[39]
Fe_3_O_4_ nanoparticles	Peroxidase-like activity for triggering extracellular matrix degradation, leading to bacterial death within acidic niches of dental-caries-causing biofilm	TEM: 213 ± 26 DLS: NA	NA	[41]
Biomimetic CoO@AuPt nanozymes	Multi-enzymatic action to produce substantial ROS without any stimuli.	TEM: 36 DLS: 61.7	−16.8	[86]
Iridium ions doped manganese ferrite nanoparticles	Iron reduced by glutathione was responsible for CDT, with synergistic magnetic hyperthermia therapy	TEM: 11.24 DLS: NA	+18.98	[88]
CeO_2_-Gd nanozyme	Superoxide dismutase mimetic activity	TEM: <10 DLS: NA	−1.61	[92]
Iron oxide nanozymes conjugated with GOx	Peroxidase-like activity of the nanozyme synergised with natural GOx activity.	TEM: 6.9 DLS: 44.5	−34.7	[93]
**Carbon-based nanozymes**				
Surface oxygenated-group enriched carbon nanotubes (o-CNTs)	High-performance peroxidase mimics for biocatalytic antibacterial therapy	TEM: 5 DLS: NA	−40	[44]
Carbon-based nanozyme doped with copper atoms and bacterial cellulose nanowhiskers	Excellent peroxidase-like activity with intelligent response to temperature. Conversion of H_2_O_2_ to ·OH radicals for killing bacteria at bio-safety levels of H_2_O_2_	DLS: 450 DLS: NA	NA	[46]
Iron doped carbon dots (Fe@Cds)	Catalysed Fenton reaction for enhanced CDT, synergized with photothermal therapy.	TEM: 77 DLS: 307	+25.8	[97]
Nitrogen-doped carbon nanospheres (N-CSs)	Peroxidase-like activity for enhanced production of cytotoxic ·OH radicals, coupled with overcoming of gemcitabine resistance by inhibiting cytidine deaminase activity by the carbon nanospheres.	TEM: 100 DLS: 250	NA	[98]
**Transition metal-chalcogenide -based nanozymes**				
Cu_2_MoS_4_ nanoplates	Near-infrared II (NIR-II) light responsive intrinsic dual (oxidase and peroxidase) enzyme-like property for highly efficient killing of bacteria	TEM NA DLS: 27.74 ± 5.92	NA	[49]
Fe_3_O_4_@MoS_2_-Ag nanozyme	Peroxidase-mimicking generation of ROS, coupled with release of Ag⁺ ions and NIR photothermal action for excellent synergistic bacterial disinfection (~100%).	TEM: ~428.9 DLS: NA	−25.0	[51]
PEGylated MoSe_2_/Au nanozyme	Peroxidase and catalase-like activity of the nanozyme, which complemented photothermal therapy (PTT) enabled by Au NPs.	TEM: 250 DLS: NA	NA	[101]
**Prussianblue- and metal-–organic-framework-based nanozyme**				
Chitosan coated prussian blue nanoparticles	Peroxidase-like activity, coupled with cationic charge on chitosan for high affinity antibacterial effect	TEM: 54.28 DLS: 60.71	+26.98	[52]
Ag^+^ ion doped prussian blue nanoparticles	Peroxidase-like activity, coupled with Ag^+^ ion release and efficient photothermal effect for potent antibacterial propensity	TEM: 40–60 DLS: 140	−15.70	[53]
Hyaluronic acid (HA) coated Ag^+^ ion loaded photosensitive metal-organic frameworks	Ag^+^ ions and generated reactive oxygen species under visible light irradiation and increased affinity to bacteria and show a strong synergistic antibacterial effect.	TEM: 85 DLS: NA	+22.4	[54]
Polydopamine (PDA) coated Yb^3+^ ions with prussian blue nanoparticles (Yb-Pb@PDA)	Enhanced Fenton reaction, aided by NIR light activation, along with scavenging of glutathione by PDA.	TEM: 250 DLS: NA	−37.9	[102]
Chlorin e6 (Ce6) loaded and hyaluronic acid (HA) tagged MIL-100 MOFs	Peroxidase-like activity of nanozyme led to generation of hydroxyl radicals and molecular oxygen via an enhanced Fenton reaction, leading to CDT and supporting Ce6-mediated PDT. HA used for cancer targeting.	SEM: 60 DLS: 138	−33	[99]
**Single atom nanozymes**				
Single iron atoms are anchored in nitrogen-doped amorphous carbon (SAF NCs)	Intrinsic peroxidase-like activity and photothermal effect in the presence of H_2_O_2_, generating abundant hydroxyl radicals for highly effective bacterial elimination	TEM: 77 DLS: 0.13 ± 0.03	NA	[64]
Atomically dispersed zinc atoms on ZIF-8 with unsaturated Zn–N_4_ sites	Peroxidase-mimicking activity mediated high antibacterial activity and wound treatment	TEM: 130 DLS: NA	NA	[65]
Single ruthenium atom incorporated on MOF (OxygeMCC-r single atom)	Catalase-like activity of nanozymes led to generation of higher oxygen for enhancing Ce6-mediated photodynamic therapy.	TEM: NA DLS: 98	NA	[100]
**Hybrid/mixed nanozymes**				
Integration of AuNPs with ultrathin graphitic carbon nitride (g-C_3_N_4_)	Superior peroxidase-activity catalyzing the decomposition of H_2_O_2_ to ·OH radicals, at bio-safety levels of H_2_O_2_ for efficient bacterial killing	TEM: 150 DLS: NA	NA	[45]
MoS_2_/rGO vertical heterostructure (VHS)	Triple enzyme-like activities (oxidase, peroxidase, and catalase) promoting free-radical generation owing to defects and photo irradiation	TEM: 10 DLS: NA	NA	[47]
Integrated nanozymes with MIL-88B (Fe) MOF surface containing Au NPs, and grafted with Ce nitrilotriacetic acid (NTA) complexes (MOF_-Au-Ce_)	Ce complexes grafted to MOF exhibits DNase-mimetic activity to catalyse hydrolysis of eDNA of biofilms. MOF doped with Au (MOF_-2.5Au_) showed enhanced peroxidase-mimetic activity with potent antibacterial activity	TEM: <200 DLS: NA	NA	[57]
Melanin coated MnO_2_ NPs with and ultrasmall Au NPs (MMF-Au).	Catalyzed both glucose oxidation and a Fenton-like reaction for improving CDT of cancer	TEM: 100 DLS: 160	−23	[85]
Self-assembled nanoparticles formulated using copper ions, carbon dots, and doxorubicin.	Catalase-like activity of both the nanozymes, coupled with doxorubicin-mediated H_2_O_2_ generation, for synergistic antitumor activity.	TEM: 76.89 DLS: 91.28	−30.3	[96]

## Data Availability

Not applicable.

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
