# Peer review of "Emerging Prospects of Nanozymes for Antibacterial and Anticancer Applications"

_biomedicines, 2022, doi:10.3390/biomedicines10061378_

Round 1

Reviewer 1 Report

The article entitled Emerging prospects of nanozymes for antibacterial and anti-cancer applications is a document of interesting subject matter.

However, it needs some major changes before being accepted. Make the following corrections:

  1. The manuscript needs to be checked again for the journal style.
  2. The abstract and conclusion are a bit too concise. Please make a general abstract and conclusion of the study.
  3. It is suggested to add the particle size, PDI, and Zeta potential related to the Tables.
  4. It is suggested to add one part on “challenges and opportunities” before conclusion part.
  5. The objective or objectives should be clearly elucidated in the last paragraph of the introduction.

Author Response

The article entitled Emerging prospects of nanozymes for antibacterial and anti-cancer applications is a document of interesting subject matter.

However, it needs some major changes before being accepted. Make the following corrections:

  1. The manuscript needs to be checked again for the journal style.

Authors: We appreciate the learned reviewer’s comment, and accordingly have carried out format checks to suit the manuscript style.

  1. The abstract and conclusion are a bit too concise. Please make a general abstract and conclusion of the study.

Authors: We have elaborated the abstract, and have included a new section prior to the Conclusions part.

  1. It is suggested to add the particle size, PDI, and Zeta potential related to the Tables.

Authors: We appreciate the reviewer’s comment, and have added the particle size (both TEM/SEM and DLS) and zeta-potential of the nanozymes (whatever are available from reported articles) in the table. Since PDI is not reported in most articles, we decided not to include this data.

  1. It is suggested to add one part on “challenges and opportunities” before conclusion part.

Authors: As suggested by the learned reviewer, we have added the “challenges and opportunities” part before the conclusions.

  1. The objective or objectives should be clearly elucidated in the last paragraph of the introduction.

Authors: We have expanded the last paragraph of Introduction to clarify the Objectives, as well as included an additional Figure to represent some key enzymes mentioned in this manuscript.

Reviewer 2 Report

  1. The outline of this review article is very poor; design a proper outline of key nanozymes under the anti-bacterial and anti-cancer applications.
  2. Make a separate heading at the end on the "safety considerations for nanozymes" that can detail its potential harm compared to the natural enzymes. 
  3. Include another heading on the "Hurdles for clinical approval of nanozymes"  detail the current standing of nanozymes and update us about the key industrial developments for the market. 

Author Response

  1. The outline of this review article is very poor; design a proper outline of key nanozymes under the anti-bacterial and anti-cancer applications.

Authors: As suggested by the learned review, we have improved the outline of the manuscript.

  1. Make a separate heading at the end on the "safety considerations for nanozymes" that can detail its potential harm compared to the natural enzymes. 

Authors: We have included a new section called “challenges and opportunities” before the conclusions, where we have briefly discussed the safety aspects of the nanozymes.

  1. Include another heading on the "Hurdles for clinical approval of nanozymes"  detail the current standing of nanozymes and update us about the key industrial developments for the market.

Authors: We have included a new section called “challenges and opportunities” before the conclusions, where we have briefly discussed the clinical aspects of the nanozymes.

Reviewer 3 Report

Indrajit Roy and coworkers appreciably reviewed the topic entitled “Emerging prospects of nanozymes for antibacterial and anti-cancer applications” with a particular focus on nanoenzyme-mediated antibacterial and anti-cancer applications.

Overall, the authors provide an appreciable review that is worth getting published in the MDPI Biomedicines, however, I recommend a detailed revision addressing the following minor issues carefully:

  1. Clearly define what are nanoenzymes?
  2. I would suggest the authors accommodate a sentence about how nanoenzymes are differentiated from nanoreactors in the introduction with the citations. The following explanation may help the readers to understand easily.                Nanoenzymes are nanoparticles exhibiting enzyme-like properties and are structurally different flow-through nanoparticles with encapsulated enzymes retaining their catalytic activities, namely nanoreactors. (https://doi.org/10.1016/j.trac.2021.116419 and https://doi.org/10.1002/anie.201706964)
  3. Define ROS after its first appearance in the manuscript. Reactive oxygen species (ROS)
  4. Write the full form of TMB  Please note that this appears only one time, so no need to abbreviate.
  5. Write full form of PtCu alloy.  Platinum-copper (PtCu) alloy
  6. exopolysaccharides (EPS) and extracellular polymeric substance (EPS). Two times abbreviated.
  7. In mice bearing 4T1 tumors,  In mice bearing 4T1 tumor
  8. The MOF behaved as the enzymes peroxidase,  in this sentence only one enzyme is catalase. Hence, the need for the plural form, enzymes.
  9. Please do give a gap here, H2O2and  H2O2 and
  10. H2O2and generate reactive oxygen species (ROS),  Here, in this sentence authors can use the abbreviation, ROS, as they will introduce the abbreviation ROS after its first appearance in the text.
  11. Caplan-Meier  Kaplan-Meier diagram
  12. Figure 5: Define what is H&E?  hematoxylin and eosin staining
  13. Define what is PNzyme-MTO?
  14. The word, tumor microenvironment (TME) was used many times (pages 9, 14, and figure 8. On page 9, it is the first time appeared. Abbreviate here as TME.

Author Response

Indrajit Roy and coworkers appreciably reviewed the topic entitled “Emerging prospects of nanozymes for antibacterial and anti-cancer applications” with a particular focus on nanoenzyme-mediated antibacterial and anti-cancer applications. Overall, the authors provide an appreciable review that is worth getting published in the MDPI Biomedicines, however, I recommend a detailed revision addressing the following minor issues carefully:

  1. Clearly define what are nanoenzymes?

Authors: We appreciate the Reviewer’s comment, and have better clarified the term nanozymes.

  1. I would suggest the authors accommodate a sentence about how nanoenzymes are differentiated from nanoreactors in the introduction with the citations. The following explanation may help the readers to understand easily.                Nanoenzymes are nanoparticles exhibiting enzyme-like properties and are structurally different from nanoparticles with encapsulated enzymes retaining their catalytic activities, namely nanoreactors. (https://doi.org/10.1016/j.trac.2021.116419 and https://doi.org/10.1002/anie.201706964)  

Authors: We thank the learned Reviewer for this very crucial suggestion, and have modified the text accordingly to reflect the difference between ‘nanozymes’ and ‘nanoreactors’.

  1. Define ROS after its first appearance in the manuscript. Reactive oxygen species (ROS) Authors: We have made this correction.

  1. Write the full form of TMB à Please note that this appears only one time, so no need to abbreviate.

             Authors: We have made this correction.

  1. Write full form of PtCu alloy. à Platinum-copper (PtCu) alloy

Authors: We have made this correction.

  1. exopolysaccharides (EPS) and extracellular polymeric substance (EPS). Two times abbreviated.

Authors: The abbreviation EPS has been used only for extracellular polymeric substance in the revised manuscript.

  1. In mice bearing 4T1 tumors, à In mice bearing 4T1 tumor

Authors: We have made this correction.

  1. The MOF behaved as the enzymes peroxidase, à in this sentence only one enzyme is catalase. Hence, the need for the plural form, enzymes.

Authors: We have made this correction.

  1. Please do give a gap here, H2O2and à H2O2 and 

Authors: We have made this correction.

  1. H2O2and generate reactive oxygen species (ROS), à Here, in this sentence authors can use the abbreviation, ROS, as they will introduce the abbreviation ROS after its first appearance in the text.

Authors: We have made this correction.

  1. Caplan-Meier à Kaplan-Meier diagram

Authors: We have made this correction.

  1. Figure 5: Define what is H&E? à hematoxylin and eosin staining

Authors: We have made this correction.

  1. Define what is PNzyme-MTO?

Authors: This term has been clarified in the revised text

  1. The word, tumor microenvironment (TME) was used many times (pages 9, 14, and figure 8. On page 9, it is the first time appeared. Abbreviate here as TME.

Authors: We have made this correction.

Round 2

Reviewer 1 Report

Authors addressed all comments carefully. 

Reviewer 2 Report

Thanks for the revisions!